# OSov: An Interactive Web Server to Evaluate Prognostic Biomarkers for Ovarian Cancer

**DOI:** 10.3390/biology11010023

**Published:** 2021-12-24

**Authors:** Zhongyi Yan, Qiang Wang, Susu Zhao, Longxiang Xie, Lu Zhang, Yali Han, Baokun Zhang, Huimin Li, Xiangqian Guo

**Affiliations:** 1Institute of Biomedical Informatics, Bioinformatics Center, School of Basic Medical Sciences, Henan University, Kaifeng 475004, China; yanzy@henu.edu.cn (Z.Y.); zhaosusue@163.com (S.Z.); 10190141@vip.henu.edu.cn (L.X.); 10190146@vip.henu.edu.cn (L.Z.); 10190150@vip.henu.edu.cn (Y.H.); 10190151@vip.henu.edu.cn (H.L.); 2School of Software, Henan University, Kaifeng 475004, China; qiangwang@henu.edu.cn; 3Department of Biotechnology, Beijing Institute of Radiation Medicine, Beijing Key Laboratory of New Molecular Diagnosis Technologies for Infectious Diseases, Beijing 100850, China; Kunbzh0201@163.com

**Keywords:** survival, ovarian cancer, biomarker, prognosis, nomogram

## Abstract

**Simple Summary:**

The OSov web server incorporates gene expression profiles with clinical risk factors to estimate the ovarian cancers patients’ survival, and provides a tool for multiple analysis, such as forest-plot, uni/multi-variate survival analysis, Kaplan-Meier plot and nomogram construction.

**Abstract:**

Ovarian cancer is one of the most aggressive and highly lethal gynecological cancers. The purpose of our study is to build a free prognostic web server to help researchers discover potential prognostic biomarkers by integrating gene expression profiling data and clinical follow-up information of ovarian cancer. We construct a prognostic web server OSov (***O***nline consensus ***S***urvival analysis for ***Ov***arian cancer) based on RNA expression profiles. OSov is a user-friendly web server which could present a Kaplan–Meier plot, forest plot, nomogram and survival summary table of queried genes in each individual cohort to evaluate the prognostic potency of each queried gene. To assess the performance of OSov web server, 163 previously published prognostic biomarkers of ovarian cancer were tested and 72% of them had their prognostic values confirmed in OSov. It is a free and valuable prognostic web server to screen and assess survival-associated biomarkers for ovarian cancer.

## 1. Introduction

As one of the most aggressive gynecological cancers with a high fatality rate, the effective screening regimen for ovarian cancer is yet to be established, and the long-term prognosis has not dramatically changed in the past 20 years [1,2]. Ovarian cancer has four histological subtypes, including serous, endometrioid, clear cell and mucinous carcinoma. Notably, high-grade serous carcinoma comprises 70% of ovarian cancers with the worst survival rate, present in older women with advanced disease (stage III or IV) and *TP53* mutations. Conversely, low-grade serous carcinoma is present in young women with a better prognosis, responding poorly to chemotherapy. Endometrioid adenocarcinoma and clear cell carcinoma representatively display histological stage I/II and are frequently related to pelvic endometriosis. Mucinous carcinoma is a fairly uncommon tumor with highly variable outcome [3]. Therefore, there is an urgent need to develop prognostic biomarkers for ovarian patients to predict clinical outcome, identify high-risk patients and guide clinical management.

Analyzing gene expression profiles of tumors with patient clinical follow-up information is a valuable way to facilitate the development of prognostic biomarkers. However, a major bottleneck for researchers with limited bioinformatics skills is how to analyze and integrate these high dimension profiling data. Additionally, previous reports showed that several bioinformatics tools can quickly measure the association between gene expression and patients’ outcomes, such as a Kaplan–Meier plot (KM plot) for ovarian cancer, containing 2190 patients from 15 cohorts in 2017 version [4]. However, the measurement of independent cross-validation with forest plot and translational function of nomogram are unfortunately lacking in KM plot, although these two functions are most important for prognostic biomarker development and future clinical application.

Herein, an online prognostic analysis tool (named OSov) for ovarian cancer was constructed, containing 3238 ovarian cancer cases with clinical follow-up data from 22 independent cohorts collected from TCGA (The Cancer Genome Atlas) and NCBI GEO databases. The OSov can perform the Kaplan–Meier plot, forest plot, uni/multi-variates Cox regression analysis and nomogram analysis to assess the prognostic value of query gene in ovarian cancer. To test the performance of OSov, more than a hundred previously reported biomarkers were checked in OSov. In summary, OSov can not only quickly assess the value of the prognostic molecular biomarkers, but also can provide the opportunities to screen potential therapeutic targets for ovarian cancer patients.

## 2. Materials and Methods

### 2.1. Data Collection of RNA Expression Profiles and Related Clinical Information of Ovarian Cancer

The datasets in OSov were collected from NCBI Gene Expression Omnibus (GEO) and The Cancer Genome Atlas (TCGA), according to the following two criteria: (1) the dataset is from primary ovarian cancer, not from other cancers or metastatic cancer; (2) the dataset has both gene expression profiles and outcome data. The risk factors were from the clinical data and long-term follow-up information. The clinicopathological features returned contained many factors based on previous studies [5,6,7,8], such as age, histological type, etc. Age is a worse predictor in ovarian cancer development and survival [7]. Histology is another key risk factor in estimating clinical outcome and clinical management, including serous, clear cells, mucinous, etc. [9]. Indeed, those clinical factors are useful information to stratify the risk of ovarian cancer patients and to estimate clinical outcomes based on the traditional therapy. Additionally, the roles of others clinical features were stated everywhere and so will no longer be enumerated one by one. The clinical data are listed in Table 1 and Table 2.

### 2.2. Construction of Prognostic Online Web Server for Ovarian Cancer

OSov was designed to estimate ovarian cancer survival as our previously prognostic web servers with minor modification [34,35,36,37,38,39]. Briefly, the OSov is deployed on Windows server system by Java and adopts Browser/Server architecture. The server side is composed of five components: (1) Raw RNA-profiles: ovarian cancer RNA-expression-profiles from TCGA and GEO databases; (2) Backend-database: The gene-expression-profiles and clinical information were stored in the “SQL Server” as a backend database; (3) Middleware: the OSov is accessed by JDBC software to link the “SQL Server” and Java ; (4) Computations: the middleware is developed by R package “Rserve” (https://CRAN.R-project.org/package=Rserve (accessed on 20 December 2021)) to connect R and Java to produce interface side. The association between gene expression and clinical survival outcomes is calculated by R packages (“survminer”, “ggplot2”and “survival”), which generate Kaplan–Meier (KM) survival curves with log-rank *p* value and calculate Hazard Ratio and 95% Confidence Intervals (HR and 95%CI); (5) User interface: The OSov employs JSP, HTML and JavaScript for front-end page to retrieve user input and displays forest-plot analysis, uni/multi-factor-survival analysis, KM plot and nomogram analysis. R package “forest-plot” is adopted to generate the forest plot for input gene in OSov. Univariate and multi-variate Cox regression analysis are used to estimate the prognostic values of the risk-factors and the query gene.

In addition, risk variables in univariate analysis (*p* < 0.2) were subject to multivariate analysis and nomogram analysis. To further build ovarian cancer patients risk model, “rms” package is applied to develop nomogram which provides visualized risk prediction based on the variables screened from univariate Cox analysis [40]. The output page can display the forest plot and survival analysis table. The subordinate page can display the KM plot figure, nomogram model and uni/multi-variate prognostic table for the input gene. All the figures could be as the “. JPEG” style. The OSov system flowchart was almost similar to our previous prognostic web server described [38,41]. The OSov can be publicly accessed at http://bioinfo.henu.edu.cn/OV/OVList.jsp (accessed on 20 December 2021).

### 2.3. Collection and Validation of Previously Published Prognostic Biomarkers of Ovarian Cancer in OSov

Previously published prognostic biomarkers were collected from NCBI PubMed to test the performance of OSov, using following keywords: “ovarian cancer” and “biomarker” and “survival” and “prognostic”. We finally collected 163 reported prognostic biomarkers according to the following criteria: (1) prognostic biomarkers of the primary ovarian cancer were identified by immunohistochemistry staining (IHC) or qPCR; (2) the association between biomarkers and clinical outcomes was significant (*p* < 0.05); (3) the full manuscript could be accessed and published in English.

### 2.4. Evaluation Potential Prognostic Biomarkers

Six prognostic biomarkers from other cancer types were reported: *TUBB6* (tubulin beta 6 class V, colorectal cancer prognostic biomarker) [42], *SFRP4* (secreted frizzled related protein 4, pancreatic cancer prognostic biomarker) [43], *NUAK1* (NUAK family kinase 1, hepatocellular carcinoma prognostic biomarker) [44], *MFAP2* (microfibril associated protein 2, gastric cancer prognostic biomarker) [45], *PLIN1* (perilipin 1, breast cancer prognostic biomarker) [46], *EFNB2* (ephrin B2, oesophageal squamous cell carcinoma prognostic biomarker) [47]. Those genes were evaluated by OSov with 25% cutoff.

### 2.5. Statistical Analysis

The association of clinicopathological factors and clinical outcomes was analyzed by GraphPad-prism 8. The Cox proportional hazards regression analysis was used to analyze the association between ovarian cancer expression profiles and prognosis based on the R package “survival”. Prognostic value was assessed by KM plot analysis and log-rank test. *p* value < 0.05 is regarded as statistically significant.

## 3. Results

### 3.1. Summary of Ovarian Cancer Cohorts in OSov

Herein, we have collected 3238 ovarian cancer cases with gene expression profiles and clinical information from TCGA (1 cohort) and NCBI GEO (21 cohorts) to develop the prognostic web server for forecasting the relationship between gene expression and ovarian cancer prognosis. The summary of clinicopathological features of total ovarian cancer cases were presented in Table 1 and Table 2, showing that 75% ovarian cancer patients are diagnosed with serous cancer, which exhibits the worst overall prognostic survival than that of other histological types (Table 1 and Figure 1A). Additionally, patients with high/advanced stages (III and IV) and grades (3 and 4) also accounted for the majority of serous histological type in ovarian cancer with worst overall survival (Figure 1A–C and Table 1) [1,48]. Age is a worse prognostic factor in ovarian cancer [49], suggesting that elder patients have the shortest overall survival than patients younger than 50 years (Figure 1D).

The utmost challenge for researchers without bioinformatics skills is that o**f** how to discover potential tumor prognostic biomarkers based on the high dimensional gene expression profiles and associated clinical factors. To overcome this challenge, we established a prognostic web server, named as ***O***nline consensus ***S***urvival for ***Ov***arian cancer (OSov), which adopts the Kaplan–Meier plot, forest plot, uni/multi-variates analysis and nomogram to explore the association of gene expression profiles and follow-up information, and eventually evaluate the prognostic value of interesting gene in ovarian cancer. In brief, OSov contains 22 cohorts and implants a set of confounding clinical factors to help users perform whole or subgroup outcome analysis for ovarian cancer. To use OSov, users first need to type an official gene symbol into the textbox, then select either a specific or all cohort(s), and choose an appropriate cutoff value of gene expression to split the ovarian cancer patients into subgroups. Additionally, by then clicking the “survival analysis” button, the home page will display a forest plot and a survival summary table for each independent cohort to quickly evaluate the prognostic abilities of the query gene in ovarian cancer. Additionally, the survival summary analysis table displays uni/multi-variates analysis results for the query gene and confounding risk factors for ovarian cancer patients, such as age, stage, histological type, etc. For special needs from certain researchers, such as subgroup analysis, users can limit the analysis in a subgroup of ovarian cancer patients with particular factors, such as race, stage, etc. OSov can also provide the nomogram for query gene and prognostic clinical factors to forecast the risk for each individual cancer patient by a pre-built risk model if multivariate Cox regression is established and a sufficient sample size is available.

### 3.2. Usage of OSov and Evaluation of Previously Published Ovarian Cancer Prognostic Biomarkers in OSov

The most valuable function of the OSov web server is providing a platform for researchers to screen, develop and validate potential prognostic biomarkers across independent ovarian cancer cohorts, which is essential and critical for the success of biomarker development. To exhibit usage and measure the performance of OSov, we collected 163 previously published prognostic biomarkers of ovarian cancer identified by IHC or qPCR to assess the function of OSov.

As a demonstration, Tan et al. reported that *CRYAB* is a poor prognostic biomarker for ovarian cancer, with higher expression in ovarian cancer tissue than normal tissue [50]. Open the OSov homepage (http://bioinfo.henu.edu.cn/OV/OVList.jsp (accessed on 20 December 2021)), type “CRYAB” into the gene symbol textbox (cutoff value: Upper25%) and press the “Survival analysis” button; then the homepage will display a forest plot (Figure 2A) and a uni/multi-variate survival summary table for *CRYAB* gene (Figure 2B). The OSov results showed that ovarian cancer cases with high expression of gene *CRYAB* were significantly associated with poor overall survival in forest plot and the survival analysis summary table (Figure 2A,B), containing seven significant cohorts (Figure 3A–G): GSE3149 [*p* = 0.0004, HR(95%CI) = 2.36 (1.47~3.81)], GSE13876 [*p* = 0.0021, HR(95%CI) = 1.49 (1.15~1.92)], GSE8841 [*p* = 0.0037, HR(95%CI) = 4.83 (1.67~13.96)], GSE9891 [*p* = 0.0086, HR(95%CI) = 1.76 (1.15~2.69)], GSE17260 [*p* = 0.0187, HR(95%CI) = 2.12 (1.13~3.96)], GSE49997 [*p* = 0.0249, HR(95%CI) = 1.87 (1.08~3.23)], and GSE32063 [*p* = 0.0323, HR(95%CI) = 2.89 (1.09~7.66)]. Furthermore, the additional multivariate analysis showed that *CRYAB* is not an independent ovarian cancer prognostic biomarker and its prognostic role may be because of the close association with other prognostic clinical factors, such as age, grade or race (Figure 2B).

The clinical features played an essential role to predict ovarian cancer patients’ survival. However, the risk weight of those factors compared with gene expression was still puzzled for clinicians in precise treatment. The risk model-nomogram as an important component of modern medicine is essential and valuable for precise decision making. By OSov, user can build nomogram for query gene and clinical meaning factors, as present in Figure 3H, I, nomogram for gene *CRYAB* in GSE2619 and GSE9891 shows similar 1-year, 3-year and 5-year survival rate. In addition, the *CRYAB* gene is less risk than that of stage or histology to predict ovarian cancer survival. The above results suggest that *CRYAB* gene could be an unfavorable prognosis biomarker for ovarian cancer, in line with the original report [50].

To measure the reliability of OSov, 163 reported ovarian cancer prognostic biomarkers were re-evaluated by OSov. Of 163 previously reported prognostic biomarkers tested by OSov, approximate 72% of reported biomarkers (117/163) were consistent with the original studies. Nevertheless, 21 of 163 biomarkers (13%) showed inconsistent prognosis; in other words, some cohorts exhibit good prognostic survival but some cohorts show poor clinical outcome, such as *PLG* gene (Appendix A and Appendix A). Unfortunately, the remaining 11% (18/163) and 4% (7/163) of ovarian cancer prognostic biomarkers have opposite and non-significant prognostic roles in OSov, respectively. Several reasons may bring about opposite and non-significant results in OSov with previous reports, such as different testing sample types or methods (*NOP14* was tested by blood using qPCR method in original study, not tissues) [51], stage (High SQSTM1/p62 protein is associated with worse prognosis in high advanced stage ovarian cancer [52], but are not significantly associated with prognosis in the whole cohort with mixed stages in OSov; however, when limiting analysis in a subgroup at an advanced stage, the *SQSTM1* gene can predict an unfavorable outcome in GSE26193 cohort by OSov), etc.

### 3.3. Excavating Potential Prognostic Biomarkers for Ovarian Cancer

The OSov could not only analyze the prognostic ability of potential biomarkers, but also help researchers excavate novel prognostic biomarkers for ovarian cancer. As an example, Mariani M et al. reported that *TUBB6* gene is overexpressed in colorectal tumor tissue and can predict poor outcome in colorectal cancer [42]. Unsurprisingly, decreased *TUBB6* expression was significantly associated with increased overall survival in five independent ovarian cancer cohorts (GSE9891, GSE30161, GSE3149, GSE13876 and GSE49997) in OSov (Figure 4A). However, the multivariate analysis showed that the prognostic role of *TUBB6* may be caused by its association with other prognostic clinicopathological features such as stage (GSE26193, GSE32062, GSE49997, GSE51088, GSE53963, GSE63885, GSE73614, GSE9891, and TCGA. Data not shown). Subsequently, the remaining prognostic biomarkers from other cancer types were also assessed by OSov, which demonstrated that these genes (*SFRP4*, *NUAK1*, *MFAP2*, *PLIN1* and *EFNB2)* exhibited good performance in predicting ovarian cancer patient outcome (Figure 4B–F). In summary, the OSov is a user-friendly web server to help researchers to mine potential prognostic biomarkers for ovarian cancer.

## 4. Discussion

Prognostic biomarkers not only can predict ovarian cancer clinical outcome, but also can stratify cancer patients with different risks and guide further clinical management. Currently, there are a couple of established prognostic web tools for ovarian cancer, such as KM plotter for ovarian cancer with 2190 samples [4] and GEPIA for ovarian cancer with 426 samples [53]. The KM plotter is a comprehensive prognostic tool for serval tumor types, such as ovarian cancer, breast cancer, etc. GEPIA is developed by Zhang lab to prognose cancer patients based on only TCGA expression profiles of dozens of cancer types, including ovarian cancer. More details of prognostic web tools could be seen in the review by Zheng H et al. [54]. However, the main limitations of those tools are that they do not have multivariate analysis adjusted for other clinical factors, lack forest plot for multi-cohorts’ analysis, and nomogram for clinical application.

In this study, we developed a prognostic analysis web server for ovarian cancer, named OSov, using 22 high-throughput expression profiles with long-term follow-up information. The OSov not only screens, evaluates and validates prognostic biomarkers for ovarian cancer, but also provides the opportunities for quick translation of biomarker candidates by providing three new functions beyond previous Kaplan–Meier plot [4,53], including: (1) forest plot, a way of visualization of gene-related clinical outcomes across multi-cohorts with HR and 95%CI; (2) multi-variates analysis, which test the independence of prognostic biomarkers with other clinical factors; (3) nomogram, which could predict the risk for each individual cancer patient by a pre-built risk model. The nomogram model not only estimates the risk values of gene expression, but also can help clinicians to predict the survival outcomes for patients based on the clinical features risk; and (4) cross-validation, the utmost advantage of each-cohort estimation could provide broad cross-validation from multi-cohorts to develop potential prognostic biomarkers.

Some of the clinical factors have been reported to play key roles in ovarian cancer prognosis [6,55]. For example, histological types of ovarian cancer are key risk factors for ovarian cancer and are used to estimate the patients’ survival, such as serous tumor showing higher risk than non-serous tumor [56]. Epidemiological investigation showed that the risk significantly increases once age is above 40 in ovarian cancer [6,56], as shown in Figure 1D. Figure 1 showed that the clinical factors are important risk factors to predict ovarian cancer patients’ survival, such as serous type associated with shortest survival time compared with the other histological types. As a result, the risk clinical factors and input gene were used for the nomogram risk model construction [40,57]. The OSov system also implanted outcome analysis functions for clinical features, including age, histology, stage, race, and others. As presented in the *CYRAB* gene nomogram model, the risk score of gene expression was less important than that of clinical factors, such as stage and histology (Figure 3H,I). The role of the clinical features can be displayed in a visual predictive nomogram, which can help clinicians to stratify the ovarian cancer patients based on the nomogram risk.

In this study, we collected hundreds of previously prognostic biomarkers for ovarian cancer to test the reliability for estimating the ovarian cancer survival. The results showed that majority biomarkers were according to original researches (Appendix A). However, some previously reported prognostic biomarkers are nonsignificant in OSov, are insignificant as well by other prognostic tools (data not shown). The mRNA-expression profiles were used in OSov while some of these reported ovarian cancer biomarkers were prognosed based on protein-level (immunohistochemical method), resulting in partial, inconsistent prognostic values.

The OSov can also help researchers discover prognostic biomarker candidates for ovarian cancer. Herein, we also tested the prognosis potency of six prognostic biomarkers from other cancer types in ovarian cancer. For example, *TUBB6* is overexpressed and associated with the poor survival in colorectal cancer [42]; Yang et al. reported *SFRP4* was highly expressed in pancreatic tumor lesions and predicted poor prognosis for pancreatic cancer patients [43]; *NUAK1* was reported as a worse prognostic biomarker for hepatocellular carcinoma [44]; *MFAP2* gene as a poor prognostic oncogene promotes motility via the MFAP2/integrin α5β1/FAK/ERK pathway in gastric cancer [45]; the silencing prognostic *PLIN1* and *EFNB2* inhibited tumorigenicity and extended patient survival in breast cancer [46] and oesophageal squamous cell carcinoma [47], respectively. The analysis of above six genes in OSov showed that they could be as potential prognostic biomarkers for ovarian cancer (Figure 4).

In this version, we utilized a quartile cutoff value to estimate ovarian cancer patients’ survival, such as upper 25% vs. other 75% cutoff, 50% cutoff, etc. OSov provides users a wide range of cutoff points and expectations to best assess survival. Indeed, ROC analysis would help us to select a better cutoff from above survival analysis; however, ROC analysis is not currently available in OSov and will be implanted in OSov in the near future. In addition, we only utilized transcriptome data to predict the ovarian cancer patients’survival. In the near future, multi-omics data will be utilized to forecast the ovarian cancer patients’ clinical outcomes.

## 5. Conclusions

OSov is a user-friendly and valuable web server for evaluating prognostic biomarkers for ovarian cancer based on independent RNA expression profiles by incorporating clinical factors. This tool displays multiple prognostic analysis results, such as forest-plot, univariate and multi-variate survival analysis, KM plot and nomogram analysis. Additionally, OSov can be easily accessed at http://bioinfo.henu.edu.cn/OV/OVList.jsp (accessed on 20 December 2021).

## Figures and Tables

**Figure 1 biology-11-00023-f001:**
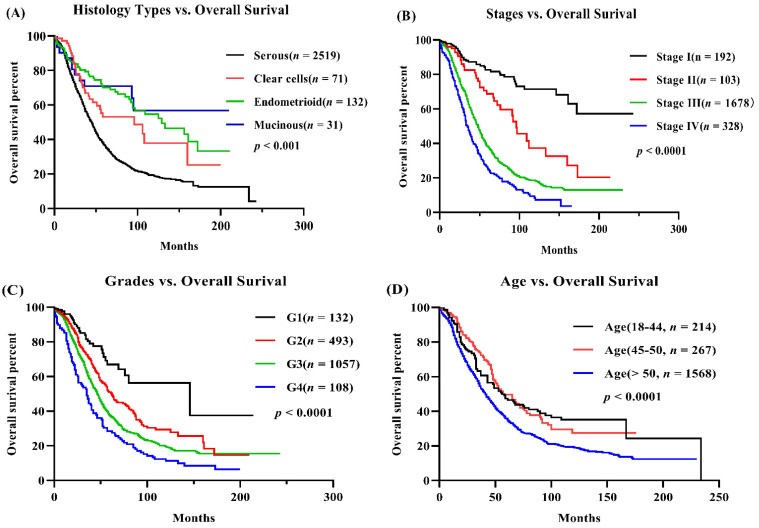
Clinico-pathological factors affect overall survival in ovarian cancer. (**A**) Histological types vs. overall survival (OS) of ovarian cancer; (**B**) Stages vs. OS; (**C**) Grades vs. OS; (**D**) Age vs. OS.

**Figure 2 biology-11-00023-f002:**
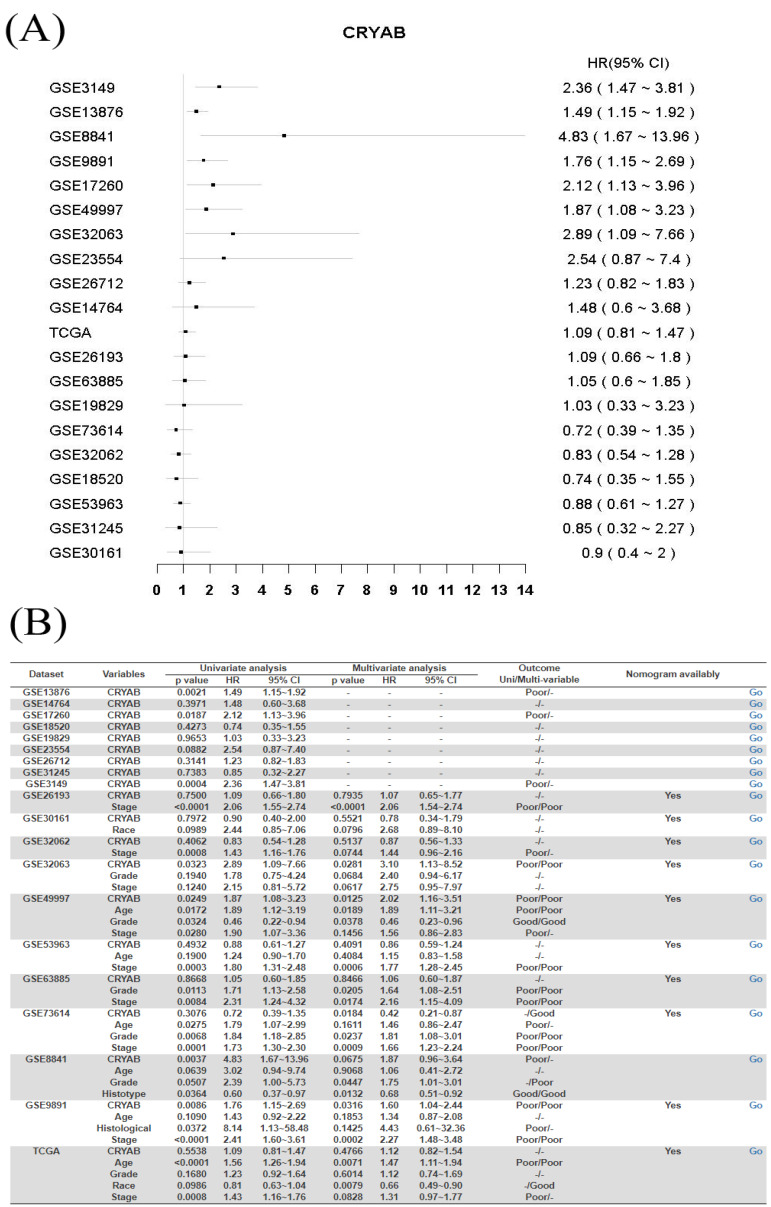
The output homepage for *CRYAB* gene survival analysis in ovarian cancer using OSov. (**A**) the forest analysis for *CRYAB* gene; (**B**) a survival analysis summary table in independent cohorts for *CRYAB* gene. CRYAB: crystallin alpha B. Cutoff value is the upper 25% vs. other 75%.

**Figure 3 biology-11-00023-f003:**
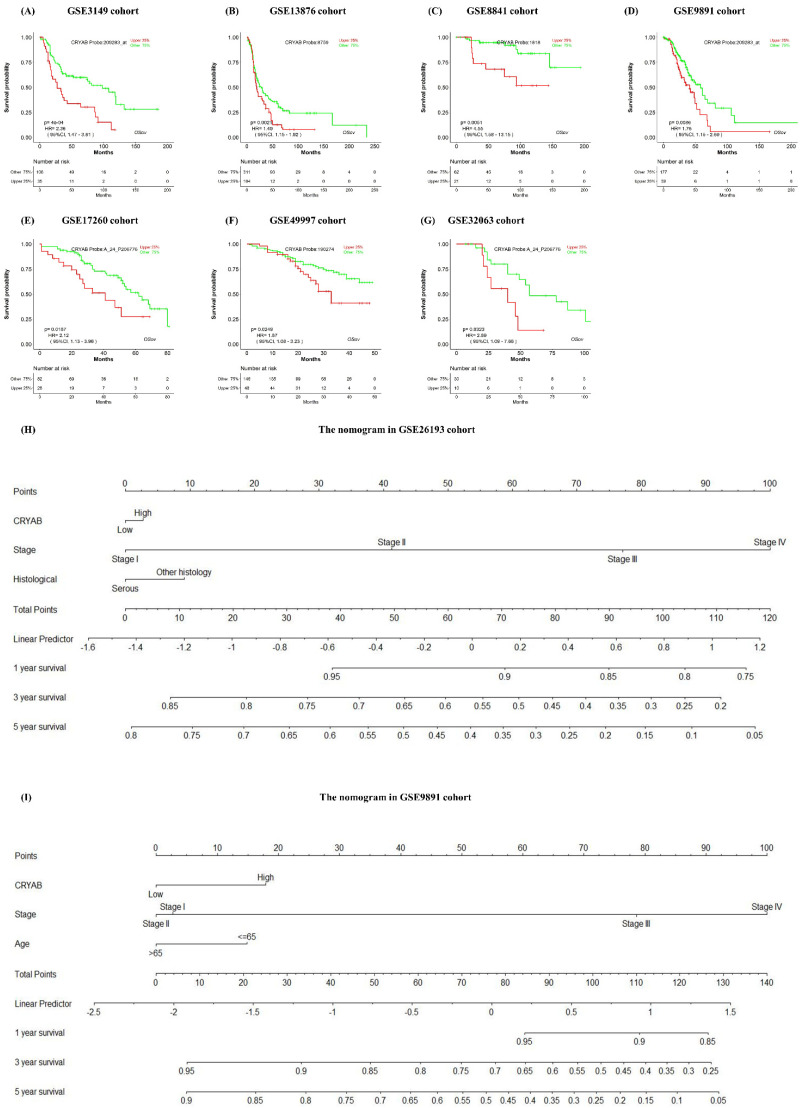
*CRYAB* is prognostic in ovarian cancer by OSov. (**A**–**G**) the overall survival (OS) analysis for *CRYAB* gene in GSE3149, GSE13876, GSE8841, GSE9891, GSE17260, GSE49997 and GSE32063 cohort, respectively; (**H**,**I**) the nomograms for *CRYAB* gene in GSE26193 and GSE9891, respectively. CRYAB: crystallin alpha B.

**Figure 4 biology-11-00023-f004:**
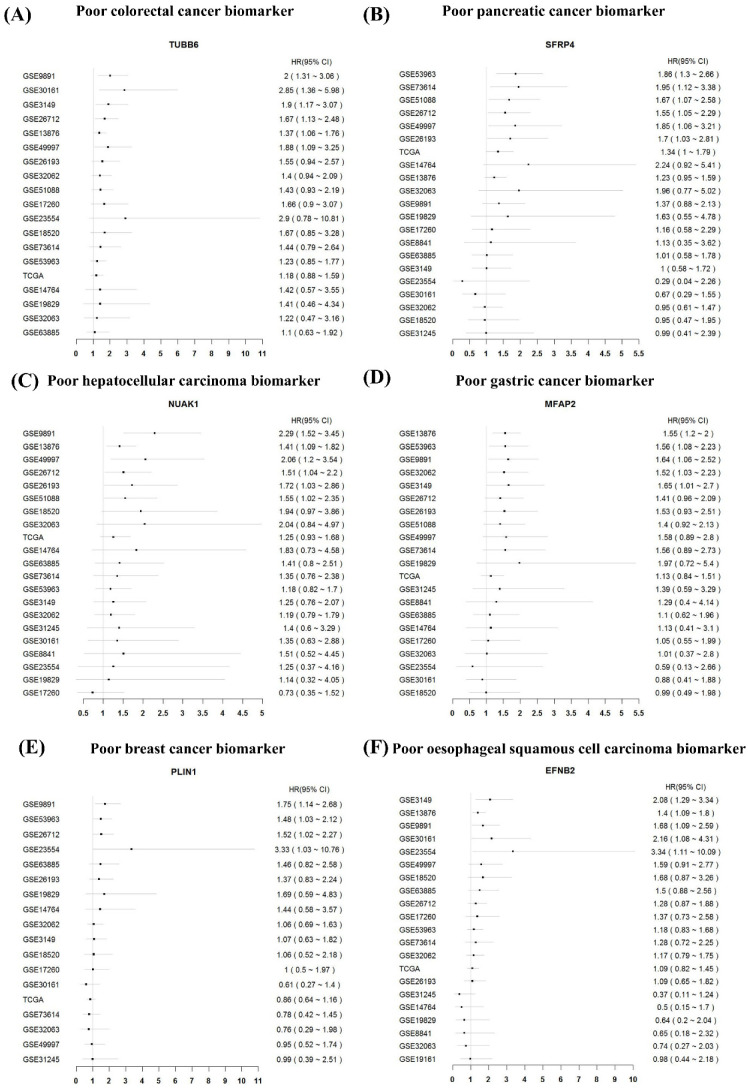
Excavation of potential prognostic biomarkers for ovarian cancer by OSov utilizing other tumor prognostic biomarkers. (**A**) *TUBB6* gene in OSov (colon cancer biomarker); (**B**) *SFRP4* gene in OSov (pancreatic cancer biomarker); (**C**) *NUAK1* gene in OSov (hepatocellular carcinoma biomarker); (**D**) *MFAP2* gene in OSov (gastric cancer biomarker); (**E**) *PLIN1* gene in OSov (breast cancer); (**F**) *EFNB2* gene in (oesophageal squamous cell carcinoma). TUBB6: Tubulin beta-6 chain; SFRP4: Secreted frizzled-related protein 4; NUAK1: NUAK family SNF1-like kinase 1; MFAP2: Microfibrillar-associated protein 2; PLIN1: Perilipin-1; EFNB2: Ephrin-B2. Cutoff: upper 25% vs. other 75%.

**Table 1 biology-11-00023-t001:** Summary of clinicopathological features of ovarian cancer in OSov.

Items	Serous Cancer (*n* = 2537)	Clear Cells Cancer (*n* = 71)	Mucinous Cancer (*n* = 31)	Endometrioid Cancer (*n* = 136)	NA * (*n* = 463)
**Age, yr ^a^**	59 (21–89)	63 (41–88)	50 (33–87)	58 (21–86)	59 (35–83)
**Stage**					
I	81	36	24	51	3
II	70	8	-	17	8
III	1571	22	7	56	40
IV	313	5	-	10	7
NA	502	-	-	2	405
**Grade**					
1	110	-	8	9	5
2	427	9	10	52	1
3	1360	54	5	68	40
4	98	3	-	4	3
NA	542	5	8	3	414
**OS, mo ^b^**	41 (1–243)	56 (2–200)	75 (3–210)	68 (1–211)	43 (1–189)
**PFS, mo**	28 (1–243)	28 (1–243)	68 (2–147)	48 (2–117)	26 (2–111)
**DFS, mo**	25 (1–115)	25 (1–115)	-	23 (2–43)	-
**DSS, mo**	40 (1–183)	40 (1–183)	-	-	47 (1–164)
**DFI, mo**	28 (1–183)	28 (1–183)	-	-	-
**PFI, mo**	22 (1–183)	22 (1–183)	-	-	-

OS: overall survival; PFS: progression-free survival; DFS: disease-free survival; DSS: disease-specific survival; DFI: disease-free interval; PFI: progression-free interval; *n*: number; NA *: Not Available; yr ^a^: year with median (quartile); mo ^b^: months with median (range).

**Table 2 biology-11-00023-t002:** Ovarian cancer cohorts in OSov.

Cohorts	Platform	Histology	Survival	*n*	Reference
GSE13876	GPL7759	SC	OS	415	[10]
GSE14764	GPL96	SC/EC/CC/UN ^#^	OS	80	[11]
GSE17260	GPL6480	SC	OS/PFS	109	[12]
GSE18520	GPL570	SC	OS	53	[13]
GSE19161	GPL9717	UN ^#^	OS	61	[14]
GSE19829	GPL8300	UN ^#^	OS/DFS	42/35	[15]
GSE23554	GPL96	SC	OS	28	[16]
GSE26193	GPL570	SC/EC/MC/CC/Other *	OS/PFS	107	[17,18]
GSE26712	GPL96	UN ^#^	DSS	186	[19,20]
GSE30161	GPL570	UN ^#^	OS/PFS	58/54	[21]
GSE31245	GPL8300	UN ^#^	OS	58	[22]
GSE3149	GPL96	UN ^#^	OS	141	[23]
GSE32062	GPL6480	SC	OS/PFS	260	[24]
GSE32063	GPL6480	SC	OS/PFS	40	[24]
GSE49997	GPL2986	SC/UN ^#^	OS/PFS	194	[25]
GSE51088	GPL7264	SC/EC/CC/MC/Other *	OS	139	[26]
GSE53963	GPL6480	SC	OS	174	[27]
GSE63885	GPL570	EC/SC/UN ^#^	OS	75	[28,29]
GSE73614	GPL6480	EC/SC/CC	OS	107	[30]
GSE8841	GPL5689	SC/EC/CC/MC/UN ^#^	OS/PFS	83	[31]
GSE9891	GPL570	SC/EC	OS/PFS	236/141	[32]
TCGA	DCC	SC	OS/DSS/DFI/PFI	582/545/286/582	[33]
Total				3238	

SC: serous; EC: endometrioid; CC: clear cells; MC: mucinous; UN ^#^: undifferentiated; OS: overall survival; PFS: progression-free survival; DFS: disease-free survival; DSS: disease-specific survival; DFI: disease-free interval; PFI: progression-free interval. Other *: other histological type, such as adenocarcinoma, Brenner tumor, carcinosarcoma or transitional cell cancer; *n*: number.

## Data Availability

All the original RNA-expression profiles and clinical information were from the publicly database, containing TCGA (https://portal.gdc.cancer.gov/ (accessed on 20 December 2021)) or PubMed GEO database (http://www.ncbi.nlm.nih.gov/geo/ (accessed on 20 December 2021)). OSov web server can be accessed at http://bioinfo.henu.edu.cn/OV/OVList.jsp (accessed on 20 December 2021). Further data of this study are available from the corresponding author upon reasonable request.

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
