# Peer review of "OSov: An Interactive Web Server to Evaluate Prognostic Biomarkers for Ovarian Cancer"

_biology, 2021, doi:10.3390/biology11010023_

Round 1
Reviewer 1 Report
Comments to the Author
Summary of paper:
In this manuscript, In this study, the authors developed a prognosis analysis web server for ovarian cancer, named OSov, using 22 high-throughput expression profiles with long-term follow-up information. The OSov is not only to screen, evaluate and validate prognostic biomarkers for ovarian cancer, but also provides the opportunities to quick translation of biomarker candidates by providing three new functions beyond Kaplan-Meier plot.
Major comments:
- The advantages of the model used in this manuscript over others in the field require further explanation.
- The description of the materials and methods is not clear enough. I suggest that the part of constructing prognosis methods should be described in more detail.
- The risk model established should be described in more detail.
- For the established model, the factors returned should be described in detail and fully after careful consideration.
- The discussion part is too simple, I think it should be described in more detail.
Author Response
Dear reviewer 1 and editor:
We here appreciate for your constructive and insightful comments and advices on our manuscript entitled “OSov: an Interactive Web Server to Evaluate Prognostic Biomarkers for Ovarian cancer” (Manuscript ID: biology-1478768). We have responded to each point and modified the text according to your insightful suggestions. The modifications in revised manuscript have been marked up using the “Track Changes” function to facilitate to view.
Summary of paper:
In this manuscript, in this study, the authors developed a prognosis analysis web server for ovarian cancer, named OSov, using 22 high-throughput expression profiles with long-term follow-up information. The OSov is not only to screen, evaluate and validate prognostic biomarkers for ovarian cancer, but also provides the opportunities to quick translation of biomarker candidates by providing three new functions beyond Kaplan-Meier plot.
Major comments:
- The advantages of the model used in this manuscript over others in the field require further explanation.
Response 1: Thanks for your insightful suggestion. There are several prognostic tools for ovarian cancer, such as KM plotter for ovarian cancer, GEPIA, etc. The KM plotter for ovarian cancer was developed by Balazs Gyorffy group based on 2,190 ovarian cancer samples from Gene Expression Omnibus(GEO) and The Cancer Genome Atlas(TCGA data) to assess the progression-free survival or overall survival. Compared to KM-plotter, the sample size of OSov tool is large, containing 3,238 ovarian cancer cases with gene expression profiles and related clinical information from TCGA (1 cohort) and NCBI GEO (21 cohorts). Second, OSov implanted multiple clinical factors to forecast survival, but KM plotter lacks this function. Third, nomogram, the goal of prognosis analysis for individual patient, is available in OSov to build a translational outcome model by inputted gene and clinical risk factors. All above main analysis are absent in GEPIA as well, especially GEPIA includes limited ovarian cancer cases, only cases from TCGA.
In order to make readers know these advantages of OSov over other tools, we have discussed and added above content into revised manuscript in lines 262-270.
2.The description of the materials and methods is not clear enough. I suggest that the part of constructing prognosis methods should be described in more detail.
Response 3: Thanks for your suggestion, more details for constructing OSov have been added into method section in lines 84-110, stating:
“OSov was designed to estimate ovarian cancer survival as our previous prognostic web servers with minor modification [34-39]. Briefly, the OSov is deployed on Windows server system by Java and adopts Browser/Server architecture. The server side is composed of five components: (1) Raw RNA-profiles: ovarian cancer RNA-expression-profiles from TCGA and GEO databases; (2) Backend-database: The gene-expression-profiles and clinical information were stored in the “SQL Server” as a backend database; (3) Middleware: the OSov is accessed by JDBC software to link the “SQL Server” and Java ; (4) Computations: the middleware is developed by R package “Rserve” (https://CRAN.R-project.org/package=Rserve) to connect R and Java to produce interface side. The association between gene expression and clinical survival outcomes is calculated by R packages (“survminer”, “ggplot2”and “survival”), which generate Kaplan-Meier (KM) curve survival curves with log-rank P value and calculate Hazard Ratio and 95% Confidence Intervals (HR and 95%CI); (5) User interface: The OSov employs JSP, HTML and JavaScript for front-end page to retrieve user input and then displays results from forest-plot analysis, uni/multi-factor-survival analysis, KM plot and nomogram analysis. R package “forest-plot” is adopted to generate the forest plot for input gene in OSov. Univariate and multi-variate Cox regression analysis are used to estimate the prognostic values of the risk-factors and the query gene.
In addition, risk variables in univariate analysis (P<0.2) were subject to multivariate analysis and nomogram analysis. To further build ovarian cancer patients risk model, “rms” package is applied to develop nomogram which provides visualized risk prediction based on the variables screened from univariate Cox analysis [40]. Those variables, including stage, age, histological type, and others, are adopted to develop nomogram model once the its P value is less than 0.2 in univariate analysis. For example, when the P value of age in univariate analysis is under 0.2, it is considered as worse risk factor, adopted to establish the nomogram model (as the Figure 3I).”
3.The risk model established should be described in more detail.
Response 2: Thanks sincerely for your insightful suggestion. The“rms” R package was used to construct the nomogram risk model by the risk variables from univariate cox analyses (P< 0.2). The risk variables include the input gene for OSov and the clinical risk factors, such as stage, age, histological type, etc. The more detailed information was described in lines 102-109 and 200-212.
4.For the established model, the factors returned should be described in detail and fully after careful consideration.
Response 4: Thanks for your insightful suggestion. The description of clinical factors returned for the risk model were added in lines 102-109, stating: “In addition, risk variables in univariate analysis (P<0.2) were subject to multivariate analysis and nomogram analysis. To further build ovarian cancer patients risk model, “rms” package is applied to develop nomogram which provides visualized risk prediction based on the variables screened from univariate Cox analysis [40]. Those variables, including stage, age, histological type, and others, are adopted to develop nomogram model once the its P value is less than 0.2 in univariate analysis. For example, when the P value of age in univariate analysis is under 0.2, it is considered as worse risk factor, adopted to establish the nomogram model (as the Figure 3I).”
In the result section, risk factors returned were also described in lines 200-212, stating: “The clinical features played an essential role to predict ovarian cancer patients’ survival. As showed in Figure 2B, multivariate analysis showed that CRYAB is not an independent ovarian cancer prognostic biomarker and its prognostic role may be because of the close association with other prognostic clinical factors, such as age, grade or race. However, the risk weight of those factors compared with gene expression was still puzzled for clinicians in precise treatment. Herein, OSov provides a visualized risk model-nomogram for query gene and clinical meaning factors on account of Cox regression analysis. As present in Fig.3H and 3I, nomogram for gene CRYAB in GSE2619 and GSE9891 shows similar 1-year, 3-year and 5-years survival rate. In addition, the CRYAB gene is less risk than that of stage or histology to predict ovarian cancer survival. The above results suggested that CRYAB gene could be as an unfavorable prognosis biomarker for ovarian cancer in line with original report [50]. OSov offers a platform to estimate the risk of clinical features and can help clinicians in personalized treatment recommendations.”
- The discussion part is too simple, I think it should be described in more detail.
Response 4: Thanks for your comment. As reviewer suggested, we have deeply discussed and added more details in the revised manuscript, as stated below:
(1)The comparison with other prognostic tools in lines 262-270:
“Currently, there are a couple of established prognostic web tools for ovarian cancer, such as KM plotter for ovarian cancer with 2,190 samples[4], GEPIA for ovarian cancer with 426 samples[53]. The KM plotter is a comprehensive prognostic tool for serval tumor types, such as ovarian cancer, breast cancer, etc. GEPIA is developed by Zhang lab to prognose cancer patients based on only TCGA expression profiles of dozens of cancer types, including ovarian cancer. More details of prognostic web tools could be seen in the review by Zheng H, et al[54]. However, the main limitations of those tools are that they don’t have multivariate analysis adjusted for other clinical factors, lack forest plot for multi-cohorts’ analysis, and nomogram for clinical application.”
(2)The clinical factor risk evaluation and nomogram analysis in lines 284-298:
“Some of the clinical factors have been reported to play key roles in ovarian cancer prognosis [6,55]. For example, histological types of ovarian cancer are key risk factors for ovarian cancer and are used to estimate the patients’ survival, such as serous tumor showing higher risk than non-serous tumor [56]. Epidemiological investigation showed that the risk significantly increases once age is above 45 in ovarian cancer [6,56], as shown in the Fig.1D. Figure 1 showed that the clinical factors are important risk factors to predict ovarian cancer patients’ survival, such as serous type associated with shortest survival time compared with the other histological types. As a result, the risk clinical factors and input gene were used for the nomogram risk model construction [40,57]. The OSov system also implanted outcome analysis functions for clinical features, including age, histology, stage, race, and others. As presented in the CYRAB gene nomogram model, the risk score of gene expression was less important than that of clinical factors, such as stage and histology (Figure3H and 3I). The role of the clinical features can be displayed in a visual predictive nomogram, which can help clinicians to stratify the ovarian cancer patients based on the nomogram risk.”
(3) Previously published biomarkers in lines 299-306:
“In this study, we collected hundreds of previously reported prognostic biomarkers for ovarian cancer to test the reliability of OSov in the ovarian cancer survival analysis. The results showed that majority biomarkers were according to original researches (Table S1). But some previously reported prognostic biomarkers are nonsignificant in OSov, are insignificant as well by other prognostic tools (data not shown). The mRNA-expression profiles were used in OSov while some of these reported ovarian-cancer-biomarkers were prognosed based on protein-level (immuno-histochemical method), resulting in partial inconsistent prognostic values.”
(4) Limitations of OSov web server in lines 319-326:
“In this version, we utilized a quartile cutoff value to estimate ovarian cancer patients’ survival, such as upper 25% vs. other 75% cutoff, 50% cutoff, etc. OSov provides users a wide range of cutoff points and expects to best assess survival. Indeed, ROC analysis would help us to select a better cut-off from above survival analysis, however, currently ROC analysis is not available in OSov yet and will be implanted in OSov in the near future. In addition, we only utilized transcriptome data to predict the ovarian cancers survival. In the near future, multi-omics data will be utilized to forecast the ovarian cancer patients’ clinical outcomes.”

Reviewer 2 Report
The manuscript entitled "OSov: an Interactive Web Server to Evaluate Prognostic Biomarkers for Ovarian cancer" describes the construction and potentials of a prognostic webserver to discover potential prognostic biomarkers by integrating gene expression profiling data and clinical follow-up information of ovarian cancer. The manuscript is well written, and the results are well described, but some questions should be addressed:
- Why use the upper 25% as a cut-off value to perform the survival analysis? Did the Authors try to use ROC analysis to define the cut-off?
- On line 30, TP53 should be italicized.
- On line 42, KM should be written out in full and, on line 78, only the abbreviation should remain.
- In Figure 1 caption “VS” should be replaced by “vs.”.
- On lines 198-201, the Authors talk about "gene cellular location". The Authors meant to talk about protein location "cytoplasmic and nuclear expression" and its relation with protein function? Please clarify.
Author Response
Dear reviewer 2 and editor:
We here appreciate for your constructive and insightful comments and advices on our manuscript entitled “OSov: an Interactive Web Server to Evaluate Prognostic Biomarkers for Ovarian cancer” (Manuscript ID: biology-1478768). We have addressed all the points you raised as described below. The modifications in revised manuscript have been marked up using the “Track Changes” function to facilitate you to view.
The manuscript entitled "OSov: an Interactive Web Server to Evaluate Prognostic Biomarkers for Ovarian cancer" describes the construction and potentials of a prognostic webserver to discover potential prognostic biomarkers by integrating gene expression profiling data and clinical follow-up information of ovarian cancer. The manuscript is well written, and the results are well described, but some questions should be addressed:
1.Why use the upper 25% as a cut-off value to perform the survival analysis? Did the Authors try to use ROC analysis to define the cut-off?
Response 1: Thanks very much for your sincere suggestion. In this vision, we used traditional quartile cutoff values to forecast ovarian cancer patients’ survival, such as upper 25% vs lower 75% cutoff, upper 50% vs lower 50% cutoff, upper 75% vs lower 25% cutoff. OSov provides users a wide range of cutoff points and expects to best assess survival. Indeed, ROC analysis would help us to select a better cut-off from above analysis, however currently ROC analysis is not available in OSov yet and will be implanted in OSov in the near future. And, above discussion has been stated in the discussion section in lines 319-324.
- On line 30, TP53 should be italicized.
Response 2: Thanks, we have italicized TP53, and checked through the whole manuscript for the gene name italics.
- On line 42, KM should be written out in full and, on line 78, only the abbreviation should remain.
Response 3: Thanks, as reviewer suggested we have used the full name Kaplan-Meier plot instead of KM plot.
- In Figure 1 caption “VS” should be replaced by “vs.”.
Response 4: Thanks, we replaced “VS” by “vs.”.
- On lines 198-201, the Authors talk about "gene cellular location". The Authors meant to talk about protein location "cytoplasmic and nuclear expression" and its relation with protein function? Please clarify.
Response 5: Thanks for your critical comment. As reviewer pointed out, our interpretation misunderstood readers, It is easy to cause misunderstanding thus, we have removed these words in the revised manuscript.

Round 2
Reviewer 1 Report
It can be accepted.